# Alternative Architecture of the *E. coli* Chemosensory Array

**DOI:** 10.3390/biom11040495

**Published:** 2021-03-25

**Authors:** Alister Burt, C. Keith Cassidy, Phillip J. Stansfeld, Irina Gutsche

**Affiliations:** 1Institut de Biologie Structurale, Université Grenoble Alpes, CEA, CNRS, IBS, 71 Avenue des Martyrs, F-38044 Grenoble, France; alister.burt@ibs.fr; 2Department of Biochemistry, University of Oxford, South Parks Road, Oxford OX1 3QU, UK; keith.cassidy@bioch.ox.ac.uk; 3Department of Chemistry, School of Life Sciences, University of Warwick, Gibbet Hill Campus, Coventry CV4 7AL, UK; phillip.stansfeld@warwick.ac.uk

**Keywords:** chemosensory array, chemotaxis, *E. coli*, cryo-electron tomography, molecular modelling

## Abstract

Chemotactic responses in motile bacteria are the result of sophisticated signal transduction by large, highly organized arrays of sensory proteins. Despite tremendous progress in the understanding of chemosensory array structure and function, a structural basis for the heightened sensitivity of networked chemoreceptors is not yet complete. Here, we present cryo-electron tomography visualisations of native-state chemosensory arrays in *E. coli* minicells. Strikingly, these arrays appear to exhibit a p2-symmetric array architecture that differs markedly from the p6-symmetric architecture previously described in *E. coli*. Based on this data, we propose molecular models of this alternative architecture and the canonical p6-symmetric assembly. We evaluate our observations and each model in the context of previously published data, assessing the functional implications of an alternative architecture and effects for future studies.

## 1. Introduction

Chemotactic responses in bacteria are mediated by large protein complexes known as chemosensory arrays, comprising thousands of copies of three primary components: transmembrane chemoreceptors (known as Methyl-accepting Chemotaxis Proteins or MCPs), the CheA histidine kinase, and the CheW coupling protein [1]⁠. Environmental cues received by the periplasmic domains of receptors initiate sensory signals that regulate CheA autophosphorylation activity, thereby modulating a cascade of intracellular phosphorylation reactions that culminate in adaptable control of the locomotor machinery [2]⁠. The highly organised clustering of chemosensory proteins integrates complex chemical signals and dramatically enhances response cooperativity, facilitating the exquisite sensitivity and behavioural adaptation characteristic of chemotactic responses [3]⁠. As such, the supramolecular array structure has been the subject of intense study, both as a model system for signal transduction and due to the involvement of chemotaxis in crucial biological processes such as cell adhesion [4]⁠, biofilm formation [4,5,6]⁠, bacterial symbiosis with plants [7]⁠ and pathogen infection of plant and human hosts [6,8,9,10]⁠.

First visualized by negative stain electron microscopy [11]⁠, the striking extended architecture of chemosensory arrays was immediately identified as an ideal target for cryo-electron microscopy [12]⁠ and cryo-electron tomography (cryo-ET) [13,14]⁠. Early cryo-ET analyses revealed that chemoreceptors in a wide range of microbial species organise as receptor trimers of dimers (ToDs) that further pack into an extended hexagonal arrangement, which is considered to be their universal feature [15,16,17]⁠. Subsequent cryo-ET studies, informed by crystal structures and molecular modelling, revealed the organisation of the baseplate region containing CheA and CheW in *E. coli* [18,19]⁠, describing the existence of six-membered (A.P5/W)_3_ rings involving the CheA P5 regulatory domain (A.P5) and CheW that interlocked the cytoplasmic tips of receptor ToDs (Figure 1). Within this organisation, pairs of ToDs are linked by a CheA dimer and two CheW monomers to form core-signalling units (CSUs), the minimal complex required for receptor-mediated CheA regulation [20,21]. The CSU associates into a p6 symmetric lattice (i.e., displaying three-, and six-fold rotational symmetry in the centers of rings and two-fold rotational symmetry at the center of every CSU). In addition, (W)_6_ rings composed exclusively of CheW, which result from the addition of a flanking CheW to each ToD of a CSU, are proposed to further interconnect the p6 lattice [19,22]⁠. Thus the general picture of chemosensory arrays that has emerged is that of an extended, pseudo-p6-symmetric lattice of interconnected CSU building blocks assembled on the inner membrane.

Recent cryo-ET and molecular dynamics studies [23,24,25]⁠ have significantly increased the understanding of intra-CSU organisation and dynamics, culminating in the structure of a complete transmembrane CSU [23]⁠. Although many questions regarding conformational rearrangements of the receptor and the kinase during signalling processes remain unanswered [2,26]⁠, even less is known about the ways in which signals are transmitted between CSUs. Generally speaking, analysis of array ultrastructure is complicated by limited long-range order in the structure, which is known to exhibit local deviations from an idealised symmetric architecture [24,27,28]⁠ and can be assembled on membranes with varying degree of local curvature. Nevertheless, characterisation of the extended architecture of the chemosensory array is an essential step towards a molecular understanding of the cooperative allosteric interactions between array components that enable its unique capacity for efficient signal integration and amplification [2,3]⁠. Here, we show that even the well-studied *E. coli* chemosensory array still holds surprises: the canonical pseudo-p6 organisation is not the only possible array architecture, nor does it adequately explain all existing experimental data. Instead, we highlight the existence of a pseudo-p2 organisation through cryo-ET observations of *E. coli* minicells. We propose molecular models of this alternate assembly as well as the canonical p6-symmetric organisation and compare their structural features in the light of current models of array structure and function.

## 2. Methods

### 2.1. WM4196 Minicell-Producing Strain Culture

WM4196 cells [23]⁠ were grown at 37 °C in L broth supplemented with 34 µg ml^−1^ chloramphenicol for 12 h. Small volumes of this culture were used to inoculate larger volumes of L broth media (without antibiotics) to an initial OD600 value of 0.075. These larger volume cultures were grown at 37 °C for 4 h until final OD600 value of 1.75. Details of the genetic characterisation, growth of the *E. coli* WM4196 strain, minicell separation from the mother cells, cryo-ET grid preparation and data acquisition are described in our previous manuscript [23].

### 2.2. Tilt Series Alignment and Tomographic Reconstruction

The acquired raw cryo-ET data, made available through the Electron Microscopy Public Image Archive (EMPIAR-10364), was reexamined in the present work. Multi-frame micrographs for each tilt image in EMPIAR-10364 were subject to whole frame alignment and image stacks were generated for each tilt-series in Warp. Tilt series were aligned automatically using the tilt-series alignment workflows available in Dynamo 1.1.478. Final bead positions from Dynamo were used to produce alignment parameters for the tilt-series with the IMOD program tiltalign, solving only for shifts and a constant tilt-axis rotation for the tilt-series with the robust fitting method. Tomograms were reconstructed based on these alignments in Warp.

### 2.3. Denoising Tomograms

Even and odd half-tomograms were generated with an isotropic voxel spacing of 5 Å using Warp [29]⁠, from even and odd frames of multi-frame micrographs, respectively. A Noise2Noise [30]⁠ based denoising convolutional neural network was trained using cryo-CARE [31]⁠. The cryo-CARE model was trained with a batch size of 16, a learning rate of 0.0004 for 200 epochs with 75 training steps per epoch. The trained network was applied to the corresponding tomogram reconstructed from the full dataset to produce a denoised tomogram.

### 2.4. Chemosensory Array Baseplate Segmentation and Visualisation

Template matching of EMD-10160 in reconstructed tomograms with a voxel spacing of 17.96 Å was performed in the Dynamo software package [32]⁠, using both in-plane and out-of-plane sampling of 12 degrees. A set of cross-correlation peaks corresponding to the CSUs in the chemosensory array with a regular organisation were observed at a distance of 25 nm from intense cross-correlation response seen for the inner-membrane. A smooth, curved surface was modelled following this set of peaks as a membrane model in Dynamo. The mesh was exported, imposing consistent normal orientations, then imported with the corresponding tomogram (voxel spacing 17.96, reconstructed using the SIRT-like filter in IMOD with 15 iterations) into Membranorama for visualisation. Given that the EMPIAR-10364 dataset contains six tilt series only, this procedure cannot be used for statistical evaluation on the relative prevalence of the p2 and p6 lattices but greatly facilitates visual examination of the array architecture.

### 2.5. Molecular Modelling

A model of the *E. coli* transmembrane CSU was constructed by extending a recent sub-nanometer resolution model of the baseplate region (PDB ID: 6S1K) [24]⁠ using the full-length *E. coli* Tsr ToD model derived in our previous manuscript [23]⁠. Flanking CheW molecules were added to both bare receptors in the CSU model using the CheW/receptor binding mode observed in PDB ID 6S1K. Extended models for both the p2 and p6 symmetries were constructed by tiling their respective unit cells along the appropriate lattice vectors. In the case of the p2 lattice, the unit cell is the CSU itself, while in the p6 lattice, the unit cell consists of three CSUs arranged within a parallelepiped as previously described [22]⁠. A lattice constant of 126 Å was used in both cases as it produced an intact baseplate and is consistent with our previous measurements [23]⁠. Modelling of the CheA.P1 and CheA.P2 domains was based on PDB ID 2LP4 [33]⁠, with missing residues in the P2-P3 linker filled in using Modeller v9.23 [34]⁠. General modelling procedures and figure renderings were conducted using VMD v1.9.4 [35]⁠.

## 3. Results and Discussion

### 3.1. A Pseudo-p6 Symmetric Array Architecture Does Not Adequately Describe All Experimental Observations

Different strategies have been employed to obtain images of chemosensory arrays with the aim of improving both their interpretability and the results of subsequent subtomogram averaging experiments. These include (i) overexpression or derepression of array and/or flagellar genes to increase array size and occurrence frequency [18,19,25]⁠, (ii) gentle cell lysis by a phage or an antibiotic to induce cytoplasmic leakage and thus reduce cell thickness [25,36,37,38]⁠, (iii) in vitro reconstitution on lipid monolayers from purified cytoplasmic components to obtain thin samples for high-resolution cryo-ET imaging [22,24]⁠, (iv) genetic manipulation of *E. coli* to express a single type of MCP, possibly with specific adaptation states or other mutations, thereby increasing array homogeneity and mimicking discrete signalling states [24,25]⁠, (v) exploration of the great variety of bacterial species [17]⁠, with often more complex and diverse chemotaxis systems, some of which are thinner than *E. coli* and (vi) use of bacterial minicells that bud near the cell poles where arrays are located [19,23]⁠. Here, we re-examine the ultastructural context of our cryo-ET volumes of the *E. coli* WM4196 minicells which led to the complete in situ CSU structure [23]⁠ (EMPIAR-101364).

Side views of the *E. coli* chemosensory arrays have a characteristic comb-like appearance with MCP “teeth” protruding from the CheA/CheW baseplate located 30 nm under the inner membrane. The lines of MCPs extend all the way into the periplasm where, in the best cases, small globular densities corresponding to periplasmic domains are visible. Whereas such brush-like shapes can be directly seen in slices perpendicular to the direction of the electron beam in the tomographic reconstruction, and often even in projection images of the minicells, the higher-order organisation is easier to infer from top views, in which the array baseplate and its hallmark honeycomb pattern is oriented perpendicular to the optical axis. We leveraged the cryo-CARE (Context-Aware image Restoration) method for Noise2Noise-based tomogram denoising [31]⁠, a technique which both improves contrast and reduces the appearance of missing wedge artefacts in tomographic reconstructions, to better visualize the chemoreceptor arrays in our low signal-to-noise tomograms of the *E. coli* WM4196 minicells (EMPIAR-101364) [23]⁠. Unexpectedly, whilst examining arrays in denoised tomograms in which receptors were aligned both perpendicular (Figure 2A) and parallel (Figure 2B) to the electron beam during imaging, we did not observe regions with an unequivocal p6-symmetry, displaying obvious 3- or 6-fold symmetry axes. Instead, they contained a repeating diamond-shaped motif arranged in a p2-symmetric fashion. Given CSU stability, biochemical necessity and the CSU reconstruction derived from these data, we postulate that the observed diamond shaped motif corresponds to a CSU (Figure 2C).

Surprised by this observation, we decided to visualize the organisation with Membranorama, a tool which allows projection of tomographic density onto an arbitrary curved 3D surface instead of simple oblique slices [39]⁠. Making use of the Dynamo software package [32]⁠, we performed template matching in the minicell tomograms using our reference array structure (EMD-10160). The resulting cross-correlation volume enabled accurate definition of a 3D surface following the intrinsic curvature of the array, onto which we projected local tomographic density from a SIRT-like filtered tomogram (see Methods). Dynamic exploration of the 3D surface, shifting the region of density projected along the surface normal, shows a pseudo-p2-symmetric assembly of CSUs in situ (Figure 2D, Appendix A). The resulting surface projections are best inspected directly in 3D (Appendix A), enabling simultaneous examination of the entire in situ array organisation in one of our *E. coli* WM4196 minicell tomograms where the pseudo-p2-symmetry is particularly evident.

Strictly speaking one should refer to a “pseudo-symmetry” when describing a 3D organisation on a curved membrane surface and use the term symmetry only for 2D lattices. However, in the remainder of this paper we will refer to the array architecture as either p2-symmetric or p6-symmetric for the sake of simplicity. It is critical to note here that the WM4196 minicells analysed in this study possess arrays with normal stoichiometries of chemosensory components, and include a native distribution of MCPs that presumably have heterogeneous adaptation states. Thus, observed structural differences cannot be attributed to the genetic manipulation of the array components.

### 3.2. Molecular Models of p2- and p6-Symmetric Array Architectures

To account for and characterise the differences between the p2- and p6-symmetric array architectures at the individual-protein level we constructed a molecular model of each lattice (Figure 3 and Figure 4, Appendix A).

To this end, we first made a model for the full-length *E. coli* CSU by extending a recent structure of the CSU baseplate (PDB ID: 6S1K) [24]⁠ and using the full-length *E. coli* Tsr receptor ToD coordinates derived in our previous manuscript [23]⁠. Models for both architectures were then constructed by tiling the CSU in accordance with the appropriate lattice vectors and using a lattice constant of 126 Å [23]⁠. As expected, both models reproduced the universal hexagonal arrangement of receptor ToDs. In addition, at the level of the kinase baseplate, the p6 model contained both the anticipated (A.P5/W)_3_ rings and empty sites for (W)_6_ rings. In contrast, the p2 model possessed only a single type of semi-formed ring whereby two CSUs provide a (A.P5/W) pair and two opposing CSUs present a bare receptor dimer. Addition of flanking CheW monomers to each CSU filled the empty (W)_6_ rings in the p6 model and gave rise to complete, two-fold symmetric (A.P5/W/W)_2_ rings in the p2 model. Thus while the flanking CheW molecules are involved in coupling neighbouring CSUs through rings in both lattices, their exact role is symmetry-dependent. In the p6 model, flanking CheWs serve to reinforce an existing lattice created by the (A.P5/W)_3_ rings formed between three CSUs, whereas in the p2 model they are essential to the formation of an extended p2 lattice, which requires the interaction of four CSUs with flanking CheWs to produce an intact baseplate. Another striking difference between the two architectures concerns the intermolecular organisation of CheA (Figure 4, Movie S2). In the p6 organisation, CheA molecules are arranged in a trimeric fashion with one monomer of each CheA dimer contributing a P5 domain to a (A.P5/W)_3_ ring in the center of the trimer, and the other monomer contributing a P5 domain to one of the three surrounding (A.P5/W)_3_ rings. These trimeric CheA arrangements are themselves organised in an interlocking hexameric fashion around central (W)_6_ rings. In the p2 organisation, however, CheA dimers form parallel stripes such that each monomer of the CheA dimer contributes its P5 domains to an opposite (A.P5/W/W)_2_ ring, resulting in chains of interlocked rings. Interestingly, this difference in CheA arrangement alters considerably both the intermolecular distance and the relative orientation of neighbouring CheA molecules, the potential consequences of which are discussed further below.

The proposed p2 architecture does not require any deviation from the current understanding of CSU structure and preserves all critical intra-CSU signalling interfaces between receptor dimers, CheA.P5 and CheW. In addition, despite the aforementioned differences in overall baseplate organisation, the same three types of interfaces are exclusively present within the baseplate rings of both lattices. These include the previously characterized interface I, involving subdomain 1 of CheA.P5 and subdomain 2 of CheW [40]⁠, and interface II, involving the subdomain 2 of CheA.P5 and subdomain 1 of CheW [41]⁠, as well as an interface involving subdomain 1 and subdomain 2 of two CheW monomers, which we term interface III (Figure 4B and Figure 5). Assuming fully interconnected p2 and p6 lattices (i.e., with all flanking CheW sites occupied), the relative abundances of ring interfaces are also identical within each lattice, namely 2× interface I, 4× interface II, and 4x interface III (Figure 5). The differences in ring structure are therefore primarily manifest as a spatial redistribution of the baseplate ring interfaces. Whilst in the p6 lattice, interfaces I and II alternate within the (A.P5/W)_3_ rings and interface III is present exclusively within the (W)_6_ rings, in the p2 lattice, all three interface types are present within each (A.P5/W/W)_2_ ring and each type is adjacent to the other two (Figure 5). Notably, the extended p2 and p6 molecular models yield no indication that interfaces I, II, or III should be subject to different structural constraints within the two lattices. For example, interface II is expected to possess interactions between roughly the same subset of interfacial residues in both the p6 and p2 lattices despite utilizing a core CheW in the former and a flanking CheW in the latter.

### 3.3. Structure Based-Analysis of Functional Implications of p2 Architecture

Given that the foremost distinguishing feature between the p2 architecture reported here and the canonical p6 architecture is their respective inter-CSU organisations, one might expect that signalling properties arising due to the interactions between CSUs would be affected. Following the elucidation of the p6 architecture in *E. coli* [18,19]⁠, structural lesions designed to affect the allosteric coupling between CSUs through disruption of interface II were shown to have dramatic effects on the cooperativity and sensitivity of the chemotactic response, suggesting that these properties were directly linked to the degree of interconnectedness between CSUs [41,42]⁠. This notion has been further advanced by a recent study investigating the detailed role of the (W)_6_ rings within the p6 architecture, showing that the cooperativity of the signalling response increases with the number and completeness of (W)_6_ rings, which vary widely depending on array assembly conditions (Piñas et al., personal communication). Given that the baseplate connectivity within the p6 lattice depends on the degree of (W)_6_ ring occupancy, one might therefore expect the p2 architecture, which necessarily exhibits a fully interconnected baseplate, to possess a higher degree of inherent cooperativity. An analysis of the number of interfaces required to get from each receptor within a given CSU to the nearest CheA in both organisations additionally suggests that signals might be more readily transmitted between neighbouring components via baseplate rings in the p2 architecture. Indeed, all three receptors in a given ToD are within two interfaces from the nearest CheA.P5 in the p2 organisation, whereas only two receptors are within this distance in the p6 architecture and the third receptor is bound to a ring that does not contain CheA at all (Figure 5). However, such analysis is complicated, especially over long distances, by the fact that specific baseplate interfaces and/or ring types could differ in flexibility and dynamics owing to their composition, which might change the efficacy of signal transmission between CSUs. An interesting corollary to the observed baseplate organisation is that signalling within the CSU itself may also be altered despite its conserved structure. Specifically, there is evidence of functional asymmetries between receptors within a ToD depending on the particular baseplate component to which they are attached [25,43]⁠. Thus the noted alterations in the structural context of each baseplate interface might cause such receptor symmetry breaking to manifest differently within the two lattices despite the conserved hexagonal arrangement of receptors (Appendix A).

Finally, differences in both the intermolecular distances and relative orientations of neighbouring CheA dimers may also have non-trivial effects on signalling and cooperativity. The P1 and P2 domains of CheA, which mediate the transfer of phosphoryl groups between CheA.P4 and the response regulator CheY, reside below the baseplate layer and are connected to each other and CheA.P3 by long, unstructured linkers. While, as far as we are aware, the possibility of inter-dimer CheA communication within chemosensory arrays has neither been proposed nor ruled out elsewhere, our models suggest that these linkers could allow interactions between the P1 domain of a given CheA and a P4 domain of multiple neighbouring CheAs (Figure 6). Such long-range CheA interactions may, therefore, contribute to cooperative array signalling and would likely be altered by the global change in CheA organisation between the p6 and p2 lattices. Ultimately, the answer to these questions will require a thorough investigation of cooperativity within arrays with different, well-defined lattice connectivity. Such an investigation should be possible through an application of the present tools to image appropriately engineered arrays in combination with emerging single-cell FRET experiments to quantify signalling responses [41,42,44]⁠.

### 3.4. Implications of the Observation of p2-Symmetric Chemosensory Arrays in E. coli

The bulk of cryo-ET imaging of chemosensory arrays in diverse biological contexts demonstrates a clear preference for the formation of a p6 symmetric architecture in *E. coli*. The question thus emerges: what are the molecular origins of the p2 symmetric architecture seen in this study? Interestingly, a recent publication by Muok et al. describes a p2 chemosensory array organisation in the spirochete *Treponema denticola* [45]⁠, which exhibits a linear CheA arrangement, including rings involving interactions between a classical CheW and a spirochete-specific CheW variant that are analogous to the (A.P5/W/W)_2_ rings seen in our model. Due to the orientation of the linear CheA strands, which always appear to run parallel to the cell axis, and because of the very high curvature of these cells perpendicular to the cell axis, the authors propose that the array organisation seen in *T. denticola* evolved specifically to accommodate the spirochetes’ high curvature. This notion is further supported by the presence of unique structural features in the CheW variant and CheA dimerisation domain, which are suggested to be critical for maintaining the structural integrity and function of these highly curved arrays. Thus, considering the *T. denticola* array organisation, it is tempting to ascribe our observation of a p2 organisation in *E. coli* minicells to their increased curvature relative to standard *E. coli* cells. However, it should be noted that the *E. coli* minicells studied here are considerably less curved than the *T. denticola* cells. Assuming an initially spherical minicell geometry (i.e., before plunge freezing), we estimate the average radius of curvature of the inner membrane to be 153 nm (Appendix A) as compared to 28 nm reported for the *T. denticola* cells. Moreover, given the well-documented stability of the chemosensory array both in vivo and in vitro [46,47,48,49]⁠, it is likely that the p2 organisation is present as such in certain WM4196 mother cells, which are necessarily less curved, prior to minicell budding. Intriguingly, re-examination of previously published data [24]⁠ in light of these new observations appears to indicate the presence of p2 architectures even in arrays of purified *E. coli* cytoplasmic components reconstituted on lipid monolayers with very low curvature (Appendix A).

One of the means of regulation of the array assembly into a p2 or p6-symmetric architecture may also originate from the assembly dynamics. The importance of the relative expression levels of array components for the formation of extended, well-ordered array patches in vitro and in situ is well documented [17,22,27]⁠. It is possible therefore that the p2 architecture may arise via an alternative assembly pathway, involving alterations in spatio-temporal regulation of component expression. Although a detailed array assembly mechanism remains elusive, the current working model in *E. coli* suggests that receptors first form ToDs that aggregate near the cellular poles, where they combine with CheA and CheW to form complete CSUs, which associate further into intermediate extended structures. While the canonical p6 architecture may accommodate either filled, partially filled or empty (W)_6_ rings at the six-fold symmetry axes of this arrangement, the p2 organisation presented here exhibits only identical nodes of (A.P5/W/W)_2_ rings. Considering that the flanking CheW molecules, which are not necessary for formation of a p6 lattice, take on a critical role in the p2 architecture, we propose an assembly pathway in which increased occupancy of these flanking positions on CSUs increases the probability of forming p2-symmetric patches. Presumably, the p2 pathway becomes more important upon increase of the local concentration of CheW during the early stages of array formation. The preponderance of the p6 architecture may therefore simply result from a more favourable assembly pathway. Interestingly, in many bacteria the CheW:CheA ratio is much higher than in *E. coli* [17]⁠. An intriguing possibility is that the p6 and p2-symmetric architectures may compete during the formation of the extended array, similar to what has been observed in some bacterial S-layers [50]⁠. Within such a phase-competition picture, it may be that intermediate-curvature settings tip the balance in favour of a p2 organisation, which becomes a structural necessity in the face of extreme curvatures, such as in *T. denticola* as suggested by Muok et al. Additional work will be required to identify how specific environmental factors contribute to array assembly and to unravel how their interplay affects array function.

As a final note, the observation of p2 patches elsewhere would imply that it may be present more broadly within previously analysed datasets, but has gone unnoticed. A possible reason for this is the use of symmetrisation during subtomogram averaging experiments. Indeed, large portions of the overall structure remain similar upon 2-, 3- and 6-fold symmetrisation (Appendix A), a property which has historically been used to improve reconstructions from small numbers of subtomograms. In an effort to enable further analysis of baseplate asymmetries, we have deposited our raw cryo-ET data for WM4196 minicells in the Electron Microscopy Data Bank (EMPIAR-10364) and would like to urge others to make available their raw data for previously published work.

In summary, we show that the p6 architecture does not adequately explain all images of *E. coli* minicells with classical chemotaxis proteins and propose an alternative which respects the observed p2 symmetry as well as the current understanding of CSU structure, including previously characterised signalling interfaces. Whilst the physiological reasons for the existence of two distinct types of array architecture with possibly differing signalling properties are as yet unknown, the discovery of alternative assemblies and their probable coexistence within collected datasets should undoubtedly stimulate further investigation and influence the way biochemical and structural data from chemotactic systems are analysed moving forward.

## 4. Reporting Summary

Further information on research design is available in the Nature Research Reporting Summary linked to this article.

## Figures and Tables

**Figure 1 biomolecules-11-00495-f001:**
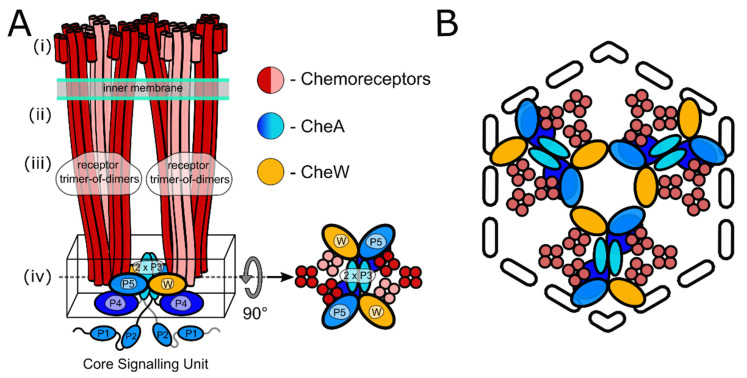
**Schematics of the core signalling unit and organisation of three CSUs into a hexagon.** (**A**) Two ToDs interact with CheA and CheW to form a CSU shown from the side. In each ToD, two MCP dimers are shown in red and one in salmon for perspective. CheA is shown in shades of blue, and CheW in gold. Domains of CheA are labelled. The baseplate region is boxed and also shown from the top. (**B**) Three CSUs assemble into a hexagon that gives rise to a (A.P5/W)_3_ ring characteristic of the pseudo-p6-symmetric array architecture (see also Figure 3B for the extended array organisation showing the formation of (W)_6_ rings).

**Figure 2 biomolecules-11-00495-f002:**
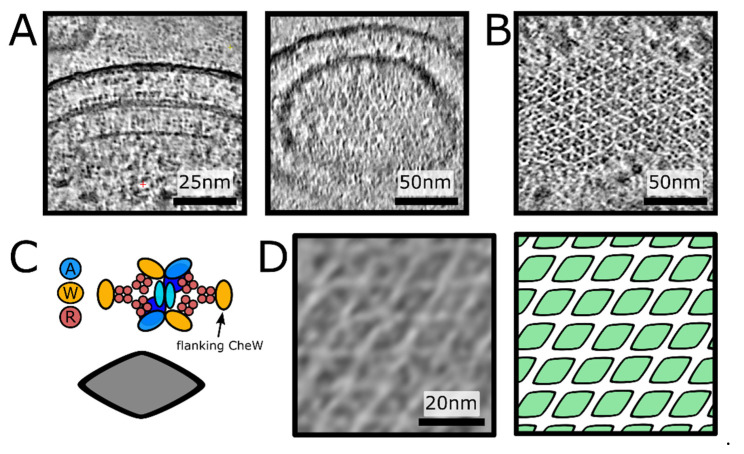
**Direct visualisation of a p2 organisation of core-signalling units in an *E. coli* minicell strain.** (**A**) 10nm thick oblique slices through a denoised tomogram with a chemoreceptor array aligned with the optical axis of the microscope. Scale bar 25 nm. (**B**) 10nm thick oblique slice through a denoised tomogram with a chemoreceptor array aligned perpendicular to the optical axis of the microscope. Scale bar 50 nm. (**C**) A schematic of the CSU (**top**) showing the positions of CheA (blue), CheW (yellow) and receptor proteins (red). A simplified visualisation of the CSU is shown as a grey diamond (**bottom**). (**D**) The chemoreceptor array from (**B**), depicted as a membranogram (**left**) following the curved surface of the array inside the cell, shows a p2 symmetric array of CSUs (**right**). The protein density in A, B and D is black. Scale bar 20 nm.

**Figure 3 biomolecules-11-00495-f003:**
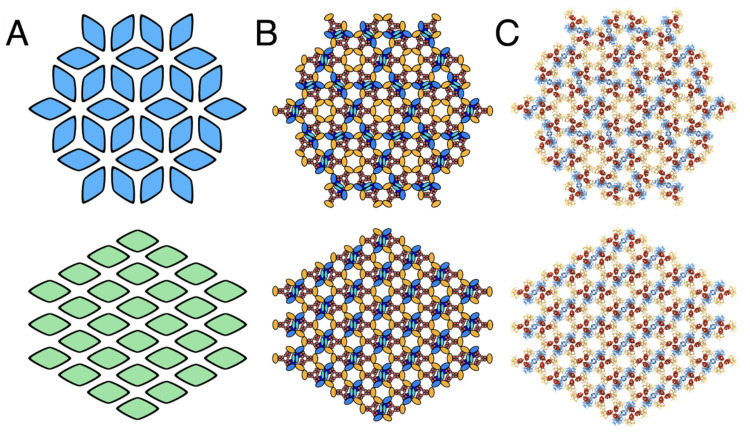
**Schematics and models of p6 and p2 chemosensory array architectures.** (**A**) Simplified schematics of p6 (**top**, blue) and p2 (**bottom**, green) array architectures with each diamond representing a CSU. (**B**) Schematics of p6 (**top**) and p2 (**bottom**) array architectures in which CheA, CheW and receptor proteins are depicted and coloured blue, green and red, respectively. (**C**) The baseplate region of all-atom models of the p6 (**top**) and p2 (**bottom**) array architectures.

**Figure 4 biomolecules-11-00495-f004:**
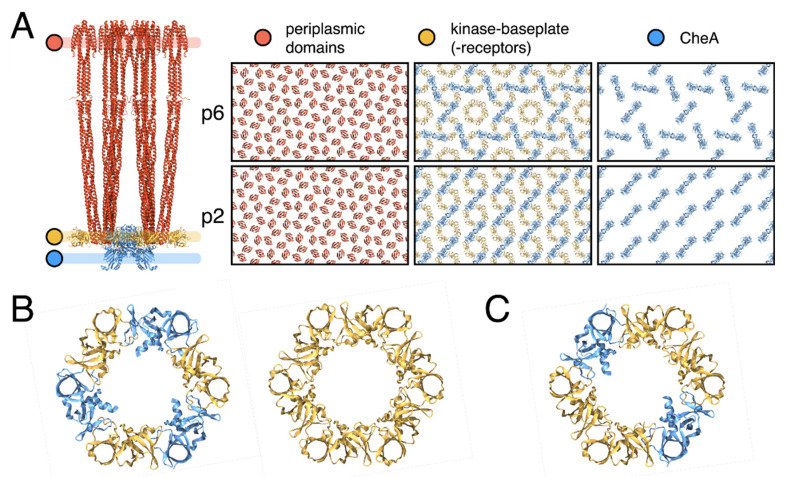
**Structural features of p6 and p2 chemosensory array architectures.** (**A**) All-atom model of the CSU (left) including flanking CheW molecules. Regions corresponding to the periplasmic domains of receptor proteins, kinase baseplate and CheA.P4 are demarcated in orange, yellow and blue, respectively. The corresponding regions in both the p6 and p2 array architectures are presented to the right, showing the near-identical receptor organisation and structural differences in the baseplate region. (**B**) The (W)_6_ and (A.P5/W)_3_ rings of CheA and CheW present in the p6 array architecture. (**C**) The (A.P5/W/W)_2_ ring of CheA and CheW present in the p2 array architecture.

**Figure 5 biomolecules-11-00495-f005:**
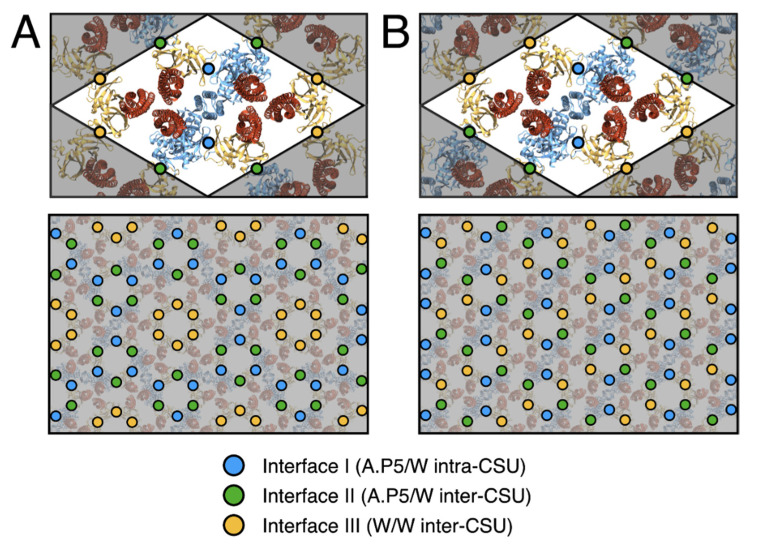
**Assembly interfaces of p6 and p2 chemosensory array architectures.** (**A**) The p6 array architecture. (**B**) The p2 array architecture. The positions of assembly interfaces I (A.P5/W intra-CSU), II (A.P5/W inter-CSU) and III (W/W inter-CSU) are depicted in blue, green and yellow, respectively. For each architecture the spatial distribution is depicted around one CSU (**top**) and a larger assembly of CSUs (**bottom**).

**Figure 6 biomolecules-11-00495-f006:**
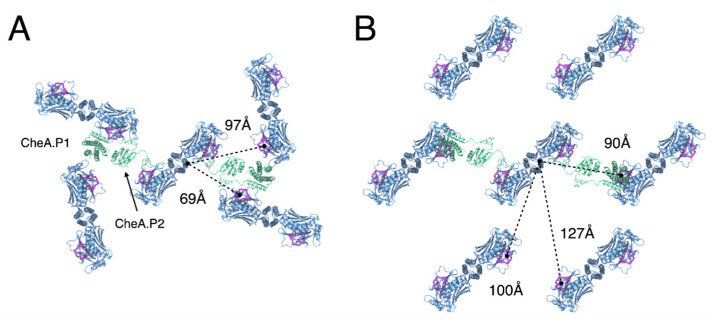
**Difference in the intermolecular distance and the relative orientation of neighbouring CheA molecules between p6 and p2 chemosensory array architectures.** (**A**) The p6 architecture. (**B**) The p2 architecture. Rough distances between a given CheA monomer, as measured from the end of the P2-P3 linker (residue I264), and ATP-binding sites on neighbouring CheAs that may be accessible in a trans-fashion by the associated P1 domain (dashed lines). The CheA.P1 and CheA.P2 domains are modelled for a centralized CheA dimer and shown in green. The ATP lid (residues 455–475) of each CheA monomer is shown in purple. Receptors, CheW, and CheA.P5 are not shown for clarity.

## Data Availability

The raw data from which tomograms were calculated, as well as reconstructed tomograms, are available on EMPIAR with accession code EMPIAR-10364. The Cryo-ET map derived from these tilt series and published in [23]⁠ is available from the EMDB with accession code EMD-10160. Coordinates for both the p2- and p6-symmetric *E. coli* array models are available for download at doi:10.5281/zenodo.4302473.

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
