# Peer review of "Alternative Architecture of the E. coli Chemosensory Array"

_biomolecules, 2021, doi:10.3390/biom11040495_

Round 1
Reviewer 1 Report
In this study the authors reanalyze one of their previously published cryo-electron tomography datasets of E.coli minicells and report a new architecture for the E.coli chemosensory array, which adopts a p2-symmetric array in addition to the previously described p6-symmetric architecture. Based on the p2-symmetric arrangement visible in their denoised cryo-electron tomograms they build a new molecular model of the chemosensory array, revealing different set of interactions between the individual chemosensory array components.
Overall, this is an interesting study with interesting implications for understanding structure and composition of bacterial chemosensory arrays. My comments are mainly focused on the technical parts of this manuscript, and how they are linked to previous studies.
Importantly, the authors should provide a more extensive comparison of the p2 and p6 architecture observed in the raw cryo-ET data. Currently, this is only done on the schematic level, but being able to directly compare the raw data would also allow readers to obtain a better understanding why the p2 architecture has been missed in previous work. Therefore, the authors should add a panel to figure 2, where an area of the chemosensory array with p6 and p2 architecture is directly compared. This is somewhat the case for Supplementary figure 4, where this is shown for in vitro data, but should be also done for the data presented in this manuscript (from EMPIAR-101364).
The authors also need to give an estimate what the ratio of p2 vs p6 architecture is within their data and also how this might have influenced their previously published results, i.e. at which resolution level they assume mixing p2 and p6 in subtomogram averaging becomes limiting. They state that the low number of tomograms in the available EMPIAR-data set limits statistical evaluation, but a comparison of p2 and p6 within these 6 tomograms should be still possible to a certain extent.
The molecular modeling is key to the results presented in this manuscript, but its description is difficult to follow, as it is not exactly clear which densities of the denoised cryo-ET data the modeling is based on. Maybe the authors can devise an additional supplemental figure where they show a fitting of both a p2 and p6 model into the data and how they deviate.
Figure 2: it would be helpful to annotate features in panel A and B to help readers who are no experts in interpreting cryo-ET data of chemosensory arrays. Specifically, in panel B the extent of the individual CSU’s could be indicated. It is also somewhat confusing that panel B and D show the same chemoreceptor array but at different magnifications. The authors should either consider adding the scale bar size directly to the figure or indicate the area that is depicted in panel D within panel B.
The central (A.P5/W)3 ring in the hexagon assembled from CSU’s appears to be threefold symmetric in Figure 1B, but perfectly 6-fold symmetric in Figure 3B. It is not entirely clear to me where this difference in the assembly arises from and if it should be representative for the p6 lattice in both cases.
The description of the orientation of the array in the main text and figure 2 appears confusing, as it reads like the opposite. When referencing panel 2A in the main text it says: “perpendicular to the electron beam (Line 116)” and in the figure legend: “aligned with the optical axis (Line 124)”. For panel 2B it reads in the main text “parallel to the electron beam” (Line 116) and in the figure legend “perpendicular to the optical axis” (Line 126). Can the authors clarify if this is correct?
The references 23 (Line 469) and 26 (Line 475) refer to the same paper (A. Burt et al, Nat. Comm 2020).
In Figure 1 the colour described for CheW is gold, in Figure 2 it is described as yellow. I would suggest keeping it consistent throughout the figures.
Reviewer 2 Report
In this paper, Burt et al. report an alternative organization for the chemoreceptor-CheA-CheW array, the well-studied sensory complex involved in environment perception during chemotactic navigation in bacteria. A single previous recent report of this organization exists, but in a very atypical organism. The authors identified this structure from cryo-electron tomography data in minicells of the main model organism for chemotaxis, E. coli. Numerous previous observations reported that CheA and CheW assemble at the base of the hexagonally arranged chemoreceptor cluster in a structure with p6-symmetry. The authors however observed a p2-symmetric arrangement of the CheA’s at the base of the chemoreceptor arrays in minicells. They additionally build a molecular model of the alternative arrangement and use it to discuss possible implications of this p2 symmetry on cooperative receptor signaling. I find the existence of this alternative receptor cluster organization extremely intriguing and important. Therefore, I think the findings of this paper are relevant to the journal, innovative and important for our understanding of cooperative chemoreceptor signaling, but I would like that the authors better substantiate their report, especially from the experimental side, before the paper could be accepted for publication.
Experimental part
- One of the main potential issues I see with the analysis of their data is the use of a single machine learning algorithm (Noise2Noise) to denoise the images of individual chemoreceptor arrays. I would like the authors to provide some controls to demonstrate that the finding of the p2-symmetry is robust, and not an artifact of the algorithm (e.g. p6-symmetry filtered out as noise by the neural network). This could involve applying alternative denoising methods and/or showing whether p6-symmetric arrays are also observed in minicells using their image analysis method (and not in a systematically biased manner, e.g. for certain orientations of the cluster relative to the instrument).
- Related to this, I think that structural analyses in Fourier space of their images (both raw, denoised and even from the membranograms) would improve the paper and strengthen the demonstration of the existence of the p2-architecture.
- At the moment there is only one relatively convincing experimental image for the p2 structure in minicells (Fig. 2B). I would suggest providing additional images taken from minicells in Supplementary. Although the authors have made their raw data available, for which they should be praised, not all readers have the means to plot them properly.
- One thing that I feel is missing is a quantitative characterization of the p2-symmetric structure. What are the lattice/primitive cell sizes? Do you see also p6-symmetric structures (see above)? What is the frequency of each symmetry being observed in the dataset? How large is the dataset, and if it is large enough, is p2 more/less frequent at higher curvature?
- Another more minor point: since the authors speculate that the formation of p2-clusters might be growth-condition dependent, they should reproduce these conditions from ref 26 in the methods (T,medium, antibiotics,etc).
- Of course, studying the occurrence of the p2 symmetry as a function of growth condition would be nice. Do the authors have any consistent data on this?
Model/simulations/discussion
Although I think the model is very neat, I have a couple minor comments about their discussion of its implications (essentially starting l.228)
- I think Supplementary Figure S1 (right panels) could be in the main text
- l.240 and following. One of the take home messages of the model in my opinion is that the p2-symmetry cannot occur at low relative levels of CheW expression. Therefore it is unfair to compare p6 with incomplete CheW rings (the apparently only viable structure at low W expression) to p2. It is not clear to me whether p6 with full W rings should be more or less cooperative than p2, and I believe that only a thorough investigation of signaling for different network connectivity could answer this questions. Such an investigation is probably beyond the scope of this paper, but that should be at least mentioned.
- On the discussion about membrane curvature: as mentioned above, providing the frequency of occurrence of p2 in minicells would help understand whether it is a relevant parameter. Did the authors manage to analyze systems with different curvatures, e.g. normal E. coli or vesicles, in a more systematic manner than just Figure S4?
- Supplementary Figure S5 is far from self-explanatory. Could you provide an actual legend to explain it better?
Round 2
Reviewer 2 Report
The authors addressed most of my concerns and questions with their response. I remain a bit skeptical of the experimental results, as it is hard to decide whether the apparent p2 symmetry is genuine or artefactual (particularly considering the power spectrum). However, the authors have provided enough data and information for the interested reader to forge their own opinion. I also do not think that additional analyses would allow this ambiguity to be alleviated, as it seems to stem from hard limits of the experiment. Since the observation is important and interesting, I think that the paper can be accepted for publication, provided a couple of minor changes are made to the manuscript:
- The information on p6 symmetry not being clearly visible in the minicells that the authors gave to the reviewers need to be more explicitly included in the text of the manuscript – that is not the case at the moment.
- I would recommend to nuance slightly the abstract - ‘Strikingly, these arrays *appear to* exhibit a p2-symmetric array architecture’ or equivalent - and to include in the discussion the nuances on the possible existence of p6 symmetry based on the averaged tomogram as discussed in the response to reviewers
Author Response
Dear Editor and Reviewers,
we are happy that our revision satisfied most of your concerns. We have now addressed the remaining comments (in red, line 19 and lines 103-104) and hope that in this revised form the manuscript will be accepted for publication in Biomolecules.
Best regards,
Irina Gutsche